# Storability and Linear Regression Models of Pericarp Browning and Decay in Fifty Litchi (*Litchi chinensis* Sonn.) Cultivars at Room Temperature Storage

**DOI:** 10.3390/foods12081725

**Published:** 2023-04-20

**Authors:** Kan Huang, Danwen Fu, Yonghua Jiang, Hailun Liu, Fachao Shi, Yingjie Wen, Changhe Cai, Jiezhen Chen, Liangxi Ou, Qian Yan

**Affiliations:** 1Institute of Fruit Tree Research, Guangdong Academy of Agricultural Sciences, Key Laboratory of South Subtropical Fruit Biology and Genetic Resource Utilization (MOA), Guangdong Province Key Laboratory of Tropical and Subtropical Fruit Tree Research, Guangzhou 510640, China; kanhuang@webmail.hzau.edu.cn (K.H.); gdglxo@126.com (L.O.); 2Institute of Nanfan & Seed Industry, Guangdong Academy of Sciences, Guangzhou 510640, China

**Keywords:** litchi, storage, physiochemical, principal component analysis, cluster analysis

## Abstract

The primary cause for the limited shelf life of litchi fruit is rapid pericarp browning and decay. This study aims to evaluate the storability of 50 litchi varieties and establish a linear regression model for pericarp browning and decay based on 11 postharvest physical and chemical indices after 9 days of storage at room temperature. The results indicated that the average value of the browning index and decay rate significantly increased to 3.29% and 63.84% of 50 litchi varieties at day 9, respectively. Different litchi varieties showed different variations in appearance indicators, quality indicators, and physiological indicators. Furthermore, principal component analysis and cluster analysis revealed that Liu Li 2 Hao exhibited the highest resistance to storage, whereas Dong Long Mi Li, Jiao Pan Li, E Dan Li 2 Hao, and Ren Shan Li were not resistant. Stepwise multiple regression analysis further demonstrated that the factors were highly correlated with the decay index, with a partial correlation coefficient of 0.437 between the effective index and the decay index. Therefore, pericarp thickness, relative conductivity, pericarp laccase activity, and total soluble solids were significant indicators for the comprehensive evaluation of litchi browning and decay, and relative conductivity was the significant determinant causing fruit browning. These findings provide a new perspective on the sustainable development of the litchi industry.

## 1. Introduction

Litchi (*Litchi chinensis* Sonn.) is a tropical fruit with high commercial value in the international fruit market [1]. As typical tropical fruit, litchi exhibits many disadvantageous after postharvest storage. The litchi fruit has a high respiration rate while being stored at room temperature, and the pericarp oxidizes rapidly within 2 to 3 days, resulting in severe quality degradation and a decline in market value [2]. The browning of litchi fruit is primarily caused by desiccation of the pericarp. For example, pericarp moisture loss usually occurs through the micro-cracks of pericarp tissues. In addition, litchi fruit internal and external quality conditions, along with postharvest losses, directly affect fruit quality parameters such as color, quality indicators, and physiological indicators [3]. However, the objective traits, including yield, quality, and storability, are quantitative and show a highly complex relationship, resulting in lower heritability. Moreover, the postharvest physiochemical indicators of fruit also exhibit different characteristics and unclear interrelationships. Based on these effects, the correlation between different indicators of litchi fruit and fruit browning can be explored to effectively screen and identify more storage litchi varieties to extend fruit life. Therefore, plant breeders seek to improve varieties with many excellent traits through litchi breeding.

In recent years, the study of the fruit browning index and decay rate during storage has mainly focused on internal and external qualities, nutrients, respiration and the pericarp browning mechanism, free radical senescence, and cell structure [4,5,6]; these factors jointly affected the storage of fruit. The values of *L*, a**, and *b** are a direct reflection of fruit browning. Additionally, total soluble solids, titratable acidity, and ascorbic acid are important characteristics that significantly affect the taste as well as the nutritional and flavor profile of litchi fruit [7]. Peroxidase (POD), laccase, and relative conductivity are the key physiological index factors that cause the browning of fruits [8]. Many traits or indicators influence the browning and decay of litchi fruits, and their correlations are very complex due to the influence of genetic characteristics, environmental factors, and the selected samples [9]. The degree of contribution of these traits or indicators to fruit browning and decay has not been further verified.

Nowadays, there are many studies designed to determine the key factors and evaluation models for fruit browning. Specifically, an evaluation model for storage has been established for lettuce [10], tomato [11], lotus root [12], avocado [13], and drop damage fruit [14]. For example, the researchers selected 17 representative traits from 39 traits by principal component and stepwise regression analyses; the 17 traits reflect 99.3% of the total variation and can be used as a comprehensive evaluation index for pear germplasm resources [15]. Breeders can increase breeding efficiency by using fewer traits to describe an accession if a relationship between phenotypic traits is revealed. However, the mentioned models, which are related to storage conditions, are rarely used to evaluate the effects of genetics on browning and decay reactions and potential shelf life. Moreover, few studies have performed linear regression analysis on index simplification and the establishment of an evaluation model for the storability of litchi fruit. At present, statistical analysis based on storability has been used in studies on mango [16,17] and winter jujube [18]. The most important point is that a linear model was applied using principal component analysis and multiple linear regression to banana browning areas and characteristic indices, indicating that it is feasible to predict banana quality based on changes in the browning area, which could be a rapid and non-destructive detection of banana quality during storage [19]. However, there are few reports regarding the application of this approach to litchi [20,21,22]. These difficulties make the comprehensive evaluation of litchi storage quality necessary. In general, the main purpose of this study is to determine the indexes related to the browning index and decay rate of 50 litchi varieties, build an evaluation model, comprehensively evaluate the storage quality of fruit, and screen out the most storable litchi varieties under normal temperature, providing guidance for actual production.

## 2. Materials and Methods

### 2.1. Material and Treatment

The experimental site is located at Fruit Tree Research Center, Guangdong Academy of Agricultural Sciences. The orchard is on a slightly hilly terrain, with consistent soil and fertilizer management. The 50 litchi germplasm resources were adult trees of 25–30 years, with uniform growth, being free from diseases and pests, and with normal flowering and fruiting every year. Commercially mature litchi fruit (85~95% maturity) were harvested at National Litchi Germplasm in Guangzhou, China, and immediately transported to the laboratory (Table 1). The fruit were selected for uniformity of size and appearance, observed to be free of blemishes and browning, cleaned, and then soaked in a 500 mg·L^−1^ iprodione solution and 500 mg·L^−1^ prochloraz solution for 1 min to eliminate potential microbes.

After air drying, the fruit were packed in sealed polyethylene film (0.02 mm thick) bags (30 fruit per bag; 15 bags of each replicate and treatment) and held in a room at 25 ± 1 °C for 9 days. Three bags of the fruit were removed for the analysis of the indicators at 0, 3, 5, 7, and 9 days of storage. The indicators measured included apparent indicators (color, browning index, decay rate, peel thickness, peel water content), quality indicators (total soluble solids, titratable acid, *Vc*), and physiological indicators (peel relative conductivity, laccase activity, POD activity). All analyses were conducted in triplicate (Figure 1).

### 2.2. Appearance Indicators

Pericarp thickness was measured using a Vernier caliper in each replication for each treatment.

Pericarp moisture content was quantified as relative mass loss by weighing (±1 mg). The pericarp of five fruits per sample was weighed before and after oven drying for 4 h at 105 °C and re-cooling in a desiccator.

The pericarp browning and decay were visually estimated using the index, as described by other researchers [23]: (1) by measuring the extent of the browning using the following scale: 0 = no browning, 1 = less than 1/4 browning, 2 = 1/4~1/3 browning, 3 = 1/3~1/2 browning, 4 = 1/2~3/4 browning and 5 = more than 3/4 browning; (2) by measuring the extent of the decay using the following scale: 0 = no visual evidence of fungi, 1 = less than 10%, 2 = 10~30%, 3 = 31~70%, and 4 = more than 70%. A browning/decay index was calculated using the following Formulas (1) and (2):(1)$$Browning\;index=\sum (\frac{browning\;scale\times percentage\;of\;fruit\;in\;ench\;class}{5})\times 100$$
(2)$$Decay\;rate=\sum (\frac{decay\;scale\;percentage\;of\;fruit\;in\;ench\;class}{5})\times 100$$

Skin color was analyzed with a Minolta Chroma Meter (CM-400, Minolta, Ramsey, NJ, USA) and expressed in the *L* a* b** color space. The three points of the fruit were selected to analyze the skin color.

### 2.3. Quality Indicators

A set of 15 fruits per replicate was homogenized and filtered. The supernatant was collected for the analysis of total soluble solids, titratable acidity, and ascorbic acid. Total soluble solids were measured with a digital refractometer [22].

Titratable acidity was determined by titration of juice with 0.1 mol·L^−1^ NaOH [24]. Specifically, pipette 2 mL of the juice into a triangular flask and titrate with 0.025 mol L^−1^ NaOH solution; 1% phenolphthalein was used as an indicator. Each experiment was repeated in triplicate at least three times. The equation given below was used to calculate the titratable acidity:(3)$$Titratable\;acidity=\frac{C\times V1\times K}{V2}$$
where *V*1 represents the volume of NaOH consumed for titration (mL); *V*2 represents the volume of juice taken for titration (mL); *C* represents the concentration of NaOH (mol L^−1^); *K* represents the conversion factor of acid.

Ascorbic acid was determined by titrating 10 g of mixed pulp sample with the standard 2,6-dichlorophenol dye [25]. Specifically, after juice extraction, 1 mL of juice was quickly drawn in a triangular flask, and 4 mL of 2% oxalic acid solution was added and titrated with 2,6-dichloroindophenol solution until pink; the endpoint was no fading within 15 s, and the volume of consumed dye was noted. The 2,6-dichloroindophenol solution was calibrated with standard *Vc*.
(4)$$T=\frac{\left(V\times V1\right)}{V2-V}$$
where *T* represents the amount of 1 mL of 2,6-dichloroindophenol solution equivalent to the ascorbic acid standard solution (mg mL^−1^); *V*1 represents the volume of standard ascorbic acid (mL); *V*2 represents the volume of 2,6-dichloroindophenol solution consumed for the titration of standard ascorbic acid (mL); *V* represents the volume of 2,6-dichloroindophenol solution consumed for the titration of the blank (mL).

The equation given below was used to calculate the *Vc* content:(5)$$Vc\left(mg\cdot 100\;{mL}^{-1}\right)=\frac{(V3-V)\times T}{V4}$$
where *V*3 represents the volume of 2,6-dichloroindophenol solution consumed during the titration of the sample solution (mL); *V*4 represents the volume of juice taken during the titration (mL).

### 2.4. Physiological Indicator

Pericarp relative conductivity was determined using other researchers [26] method. Pericarp discs with a 5 mm diameter were removed from 10 fruit at the equatorial region (three discs per fruit), then rinsed and incubated in 25 mL of distilled water for 30 min. Next, initial electrolyte leakage was monitored with a conductivity meter (DDS-11A, Shanghai, China). Total electrolyte leakage was determined after boiling for 30 min and cooling to 25 ± 1 °C. Relative conductivity was expressed as a proportion of total electrolyte leakage.

Litchi peroxidase activity (POD) was extracted by homogenizing the pericarp (1 g) with 5 mL of 0.05 mol·L^−1^ phosphate buffer (pH 7.0). The homogenate was centrifuged for 20 min at 13,000× *g* and 4 °C, and the supernatant was collected as the crude enzyme extract. POD activity was assayed using the method as follows: in a 3 mL reaction mixture that contained 0.05 mL enzyme extract, 2.75 mL 0.05 mol·L^−1^ phosphate buffer (pH 7.0), 0.1 mL of 0.46% H_2_O_2_, and 0.1 mL of 4% guaiacol. The increase in absorbance at 470 nm due to guaiacol oxidation was recorded for 2 min using a spectrophotometer (Z 2000, HITACHI, Japan). One unit of enzyme activity was defined as the quantity of the enzyme that caused a change of 0.001 in absorbance per minute.

Litchi laccase activity was extracted by homogenizing the pericarp (8 g) with 20 mL of 0.05 mol·L^−1^ phosphate buffer (pH = 7.0) with Polyclar AT (insoluble polyvinylpyrrolidone; 10% of pericarp by weight). The homogenate was centrifuged for 20 min at 13,000× *g* and 4 °C. Then, the supernatant was collected as the crude enzyme extract. In a 3 mL reaction mixture that contained 0.05 mL enzyme extract, 2.35 mL MUB (modified universal buffer) (pH 4.0), and 0.6 mL of 0.02 M ABTS (2,2′-azinobis-(3-ethylbenzthiazoline-6-sulphonate)). The increase in absorbance at 420 nm was recorded for 2 min using a spectrophotometer (Z 2000, HITACHI, Tokyo, Japan). One unit of enzyme activity was defined as the quantity of the enzyme that caused a change of 0.001 in absorbance per minute.

### 2.5. Statistical Analysis

All the treatments and measurements were set up in a completely randomized factorial design with three replicates. All data were subjected to analysis of variance (ANOVA) with SPSS 27.0 (IBM SPSS 27; SPSS Inc., Armonk, NY, USA). The samples consisted of fifty litchi varieties under various storage times at room temperature. The eleven indexes under the various storage times were the variables. Correlation analysis, factor analysis, multiple linear stepwise regression, and clustering analysis were used as the analysis method. Factor analysis was performed using the Kaise standardized orthogonal rotation method. The variable load values in the rotating component matrix were used to select the explanation of the variables with the common factor. In addition, data were plotted using ORIGIN 2022 (Origin Laboratories, Northampton, MA, USA) software.

## 3. Results

### 3.1. The Change in Pericarp Browning and Apparent Indicators of Harvested Litchi Fruit

One of the major factors that affect the commercial value of litchi fruit is pericarp browning [27]. As storage time increases, the browning index of the fruits significantly increases, resulting in the fruits turning brown (Figure 2c). Therefore, a colorimeter was used to measure the rind color of the litchi fruit after harvest (Figure 2e–g). The *L** and *b** values showed no difference between day 0 and day 3; however, the fruit values began to decrease significantly by the fifth day, and the *L** and *b** values decreased by 20.84% and 27.62%, respectively, on the ninth day. On the other hand, the *a** value did not significantly differ within the first 5 days and then began to decrease significantly by the seventh day, and the lowest value was reached on the ninth day, with a decrease of 19.76%. It is worth noting that the *a** value represents the degree of redness of the Litchi fruit [23], which suggests that the fruit’s redness can be effectively maintained within the first 5 days after harvest.

To further evaluate the postharvest phenological characteristics of litchi fruits during storage at room temperature, the pericarp thickness (Figure 2a), moisture content (Figure 2b), and decay rate (Figure 2d) were measured. As the storage time prolonged, the pericarp thickness of litchi fruits significantly reduced, and the decay rate of litchi fruit began to increase significantly by the fifth day and decreased by 17.89% by the ninth day, while the moisture content of the fruit remained mostly unchanged. Overall, these results suggest that the best storage period for litchi fruit is within 5 days after harvesting at room temperature.

### 3.2. The Change in Quality Indicators of Harvested Litchi Fruit

This study aims to evaluate the quality of litchi fruits stored at room temperature for different time periods by determining total soluble solids, *Vc* content, and titratable acidity. As the storage time increased, a significant decrease in total soluble solids was observed in litchi fruits, namely, they decreased by 14.38% compared with day 0 (Figure 3a). The *Vc* content decreased, but the change was not significant (Figure 3b). However, the titratable acidity significantly decreased on the third day (decreased by 36.35%) and then remained stable from day 3 to day 7 and increased significantly on the ninth day (Figure 3c).

### 3.3. The Change in Physiological Indicators of Harvested Litchi Fruit

The relative conductivity of the fruit pericarp represents the cell membrane permeability or degree of cell membrane injury, and its value increases with the degree of cell aging and damage [28]. The research results indicated that there is no difference in relative conductivity within the first 5 days of room temperature storage, a significant increase was observed starting from day 7, and the maximum value was reached at day 9, with an increase of 52.22% compared to day 0 (Figure 4), indicating that the integrity of litchi fruit pericarp cell membranes can be well maintained within 5 days after harvest. Laccase is a degradation enzyme for epicatechin-mediated anthocyanin in litchi fruit tissue [29], the activity of which directly affects fruit pericarp browning. The results of this study show that laccase activity remained stable within the first 5 days of room temperature storage, with no significant upward trend, and a significant increase in laccase activity was observed on the seventh and ninth days, by 25.60% and 34.29%, respectively (Figure 4), indicating that the epicatechin-mediated anthocyanin in the fruit tissue began to degrade significantly on the 7th day, which increased the browning index of the litchi fruit pericarp. In order to further evaluate changes in the antioxidant system, this study has measured the activity of peroxidase (POD) in litchi fruit, which can effectively remove peroxides. Surprisingly, the activity of POD did not show significant changes during the room temperature storage of litchi fruit, possibly due to the fact that the antioxidant system is easily influenced by multiple factors and is itself a complex nonlinear system.

### 3.4. Correlation Analysis of Indicators at Room Temperature Storage

In this research, a correlation analysis was conducted on various indices associated with the browning index and decay rate of litchi fruits, including pericarp thickness, moisture content, total soluble solids, *Vc*, relative conductivity, *L**, *a**, *b**, and POD activity. The results revealed significant correlations between the browning index and decay rate and physicochemical indicators during room temperature storage, as shown in Figure 5. Specifically, the browning index and decay rate exhibited highly significant positive correlations with relative conductivity and laccase activity. Moreover, the browning index and decay rate showed highly significant negative correlations with pericarp thickness, total soluble solids, *Vc*, *L**, and *a**, which is the most visual response to fruit browning, indicating the result obtained was reliable. In addition, *L**, *a**, and *b** showed a significant negative correlation with relative conductivity. Therefore, the factors that significantly affected the browning index and decay rate of litchi fruit were initially screened by correlation analysis, and the potential association of these indicators with the browning index and decay rate was initially verified to lay the foundation for the subsequent analysis.

### 3.5. Principal Component Analysis and Cluster Analysis

Moisture content, titratable acidity, *b**, and POD were excluded from the analysis due to variance values below 0.6. The remaining seven physicochemical indicators were considered for PCA analysis, with a measure of sample adequacy greater than 0.8. The five principal components, with eigenvalues >0.7, accounted for 85.21% of the total variance among the accessions (Table 2). PC1, which explained 30.375% of the total variance, was primarily influenced by pericarp thickness and total soluble solids. PC2, which accounted for 20.92% of the variance, was primarily associated with *Vc* and *a**. The eleven variables were categorized into five principal factors based on high loads, and these five factors provided plausible explanations for the browning and decay metabolism of litchi fruit. The following model was developed based on principal component analysis:(6)$$F_{Browning\&Rotting}=0.2457F_{1}+0.1888F_{2}+0.1749F_{3}+0.1317F_{4}+0.111F_{5}$$

Total scores for browning and rotting were calculated using the indicators obtained at 0, 3, 5, 7, and 9 days, and the correlations of these indicators with the browning index and decay rate were analyzed (Table 3). Cluster analysis was performed based on *F*_Browning&Rotting_, and the 50 litchi accessions were categorized into five main clusters based on their storability characteristics. Main Cluster 1 consisted of a single accession with the highest storability, while Main Cluster 2 comprised 13 accessions with relatively better storability. Main Cluster 3 included 17 accessions with good storability, whereas Main Cluster 4 encompassed 14 accessions with poor storability. Finally, Main Cluster 5 comprised five accessions with the lowest storability. Therefore, based on principal component analysis and cluster analysis, we identified and selected the most storable lychee cultivar, Liu Li 2 Hao, as well as relatively storable cultivars Wu Jun 2 Hao, Shang Shu Huai, and Qiong Shan 31 Hao, Zu Shan 4 Hao, Jian Ye Li, Jing Xing, Hai Guo 4 Hao, Xue Huai Zi, Jia Yuan Mi Li, Zhuang Yuan 1 Hao, Cheng Tuo, Huai Zhi, and Yuan Duan Li, from a pool of 50 lychee cultivars stored at room temperature. Additionally, we identified the least storable cultivars, which include An Duo Ji Dan, Ren Shan Li, and E Dan Li 2 Hao, Jiao Pan Li, and Dong Long Mi Li.

### 3.6. Stepwise Multiple Regression Analysis

A summary of the browning index model showed that the R^2^ value was 0.669, indicating that 66.9% (R = 0.823) of the difference in response variables is attributable to the change in control variables. According to the results above, the relevant indices of litchi fruit, such as relative conductivity, total soluble solids, pericarp thickness, laccase, *L**, *a**, and *Vc*, were found to be major factors contributing to changes in the browning index. Therefore, a stepwise regression procedure involving seven steps was employed to select the controlled variables (Table 4). The results indicated that step 7, which included the controlled variables relative conductivity, total soluble solids, pericarp thickness, laccase, *L*, a**, and *Vc*, achieved the best fit to the data. Thus, the final model included seven controlled variables, and based on these variables, the regression model was constructed:(7)$$Y=5.184+4.905X_{1}-0.28X_{2}-0.924X_{3}+3.25X_{4}-0.004X_{5}-0.026X_{6}-2.244X_{7}$$

After performing a *t*-test, the results showed that the standardized coefficients for relative conductivity, total soluble solids, pericarp thickness, laccase activity, *L**, *a**, and *Vc* were statistically significant. These factors were considered to be highly correlated with the browning index, which was found to have a partial correlation coefficient of 0.318. The order of these factors in terms of absolute values, from highest to lowest, was as follows: relative conductivity > laccase activity > *Vc* > pericarp thickness > total soluble solids > *a** value > *L** value. These findings suggest that lower relative conductivity and lower laccase activity, as well as higher values of pericarp thickness, *Vc*, *a**, and *L**, contribute to a reduction in the browning index. Moreover, the theoretical browning index calculated by the model exhibited a positive correlation with the measured value (R = 0.886 **), which indicates a high degree of agreement between the theoretical and measured values.

The summary of the disease index model revealed an R^2^ value of 0.573, indicating that there were 57.3% (R = 0.757) changes in the response variable. Four controlled variables were identified in the final model, including relative conductivity, total soluble solids, pericarp thickness, and laccase activity; based on these variables, a regression model was constructed, and four steps were taken to select the controlled variables, as presented in Table 4. The statistical procedures indicated that step 4, which included the controlled variables of relative conductivity, total soluble solids, pericarp thickness, and laccase activity, resulted in a better fit to the data. Consequently, the final model included the four controlled variables, and the regression model was constructed accordingly:(8)$$Y=0.861+1.069X_{1}-0.059X_{2}-0.216X_{3}+0.493X_{4}$$

After conducting a *t*-test, it was shown that the standardized coefficients for relative conductivity, total soluble solids, pericarp thickness, and laccase activity were statistically significant. Further analysis of the data revealed that these factors were highly correlated with the decay index, with a partial correlation coefficient of 0.437 between the effective index and the decay index. The order of absolute values for these factors, from highest to lowest, was as follows: relative conductivity > laccase activity > pericarp thickness > total soluble solids. These results suggested that a lower relative conductivity and lower laccase activity, in addition to higher values of pericarp thickness and total soluble solids, decreased the decay index. Moreover, the theoretical decay index calculated by the model showed a positive correlation with the measured value (R = 0.923 **), indicating a high degree of consistency between the theoretical and measured values. These seven phenotypic traits can be used as comprehensive evaluation indices for predicting the browning index and decay rate.

## 4. Discussions

### 4.1. Differences in Storability in Litchi Varieties

This study has evaluated the storage performance of 50 litchi cultivars at room temperature and further screened out litchi cultivars with better storage tolerance. The browning index and decay rate of the fruit were the most direct indicators to evaluate storage performance. As expected, with the extension of room temperature storage time, the browning index and decay rate of all cultivars increased significantly (Figure 6). The apparent indicators (such as pericarp thickness, moisture content, *L**, *a**, and *b**), quality indicators (such as total soluble solids, *Vc*, and titratable acid), and physiological indicators (such as relative conductivity, laccase activity, and POD activity) of different cultivars also showed varying degrees of change (Figure 2, Figure 3 and Figure 4). From the apparent indices of 50 litchi varieties measured at different time periods, we can draw the conclusion that the best storage period is 5 days at normal temperature after harvest. However, because of varietal differences, the 50 litchi varieties will likely exhibit significant differences in browning and decay. Based on the results, the most obvious finding to emerge from the principal component and cluster analysis was that Liu Li 2 Hao exhibited the strongest resistance to browning and decay at room temperature and can be considered the variety with the strongest storability among the 50 litchi varieties (Table 2 and Table 3). In comparison with other researchers’ results, which used postharvest treatments such as ascorbic acid treatment [30], melatonin treatment [31], and apple polyphenol treatment [32], the Liu Li 2 Hao variety still exhibited a lower browning index and better storage performance during the same time period. Moreover, the results from the stepwise multiple regression analysis indicated that the strength of Liu Li 2 Hao lies in the fact that it can maintain a higher level of total soluble solids and pericarp thickness and lower relative conductivity during room temperature storage. In addition, Dong Long Mi Li showed the worst storability at room temperature. The primary reason for this discrepancy may be attributed to its lower level of pericarp thickness and higher relative conductivity (Figure 2 and Figure 4). The classification results of the different litchi fruit varieties obtained in this study were generally consistent with the conventional understanding of their storability. As a result, these findings could serve as a foundation for selecting suitable litchi breeding parents.

### 4.2. Impact Factors for Pericarp Browning

Under the conditions of postharvest storage at room temperature, in addition to the apparent indicators of fruit, the quality and corresponding physiological changes of the fruit should be the secondary key indicators that need to be focused on. The overall trend of total soluble solids in the fruit of 50 litchi varieties showed a significant decrease, indicating a decrease in edible quality [33]. The titratable acidity of litchi fruits also showed a similar trend, mainly due to the aging and oxidation reactions of the fruit during postharvest storage [34,35]. The accumulation of reactive oxygen species (ROS) during fruit storage reduces the storage quality of litchi fruit. *Vc*, usually as an antioxidant, can prevent plants from oxidative damage [36]. The significant increase in *Vc* content in litchi fruit on the ninth day may be related to the plant’s self-regulation mechanism. The above indicators affected the color of the litchi pericarp to a great extent, thus affecting the commercial value.

Furthermore, the accumulation of ROS depends on the plant’s antioxidant system [37]. As a result, POD has the ability to effectively regulate ROS concentrations, preventing ROS damage [38,39]. Despite its exclusion through component analysis and clustering analysis, its role in antioxidant defense is worth discussing. The results of this study suggest that the activity of POD in litchi fruit remains stable and does not show a significant increase or decrease with prolonged storage time at room temperature, indicating that litchi fruit tissue maintains a certain degree of antioxidant activity at normal temperature and does not lose its protein function, which also contributes to the prolongation of litchi storage time. Additionally, the color of the litchi fruit pericarp is due to the abundant anthocyanins present [40]. Polyphenol oxidase (PPO) is an enzyme highly correlated with the browning of fruit pericarp, which forms brown polymers through the oxidation of phenolic compounds, flavonoids, and ascorbic acid substrates in the pericarp, resulting in pericarp browning [40]. In addition, it is noteworthy that compared with PPO, laccase has a wider substrate range [41] and stronger oxidation characteristics for phenolic substances [40]. The results showed that the laccase activity gradually increased under normal temperature storage conditions (Figure 4), which kept the anthocyanin in plant tissue at a low level and caused the litchi fruit browning. In addition, the integrity of the fruit tissue cell membrane system is closely related to its browning degree and decay rate [42]. In this study, the relative conductivity of 50 litchi varieties increased significantly from the seventh day (Figure 4), resulting in the severe browning of litchi fruit at that time (Figure 2). This result may be explained by the fact that the damage of cell membranes during the postharvest storage of litchi fruit may lead to the direct contact of enzymes and phenolic substrates to produce brown or brown substances, causing fruit browning. A stepwise multiple regression analysis can also confirm these discussions. Based on the results, the following conclusion can be drawn: among the 50 litchi varieties, Liu Li 2 Hao has the best storage capacity, mainly due to lower relative conductivity and lower POD and laccase activities.

### 4.3. Evaluation Model for Pericarp Browning

In this study, principal component analysis was used to condense a vast number of interrelated traits into a limited number of principal factor trait groups to replicate the correlation between the primary traits and the principal factor groups [43,44,45]. In summary, the results show that relative conductivity, pericarp thickness, and fruit quality are responsible for the browning of litchi fruit at room temperature and all changes that could cause the browning or decay of the pericarp. Specifically on the index of relative conductivity, the result showed that the pericarp relative conductivity had the strongest positive correlation with the fruit browning index and decay rate (Figure 5) and had the highest contribution rate to fruit browning and decay (Table 4). These results were in accord with recent studies indicating that the maintenance of membrane integrity is the key issue in prolonging the shelf life of litchi fruit [6]. After constructing a browning index evaluation model using the stepwise regression method, all predictors in the model were found to be highly representative and originating from different common factors. There was high credibility in the calculated theoretical values, which were consistent with the measured values. The effective index selected to explain browning had a high credibility of 82%. According to linear regression analysis, the higher levels of total soluble solids, pericarp thickness, *L**, *a**, and *Vc*, as well as lower relative conductivity and lower laccase activity, were conducive to improving fruit browning resistance.

### 4.4. The Application of the Linear Regression Models

The linear regression models in this study presented several advantages. In this study, the indicators of the evaluation models were the conventional physical and chemical indices of litchi storage, which was simple and easy to obtain, and the features extracted from the evaluation model were conducive to simplifying the types of indicators. The seven traits obtained from the integration and screening by principal component analysis, cluster analysis, and stepwise regression analysis can reflect more information and can be used as the main form of future litchi browning evaluation. The greatest strength of this model is that the models more accurately evaluate the storability of the fruit and avoid errors caused by subjective evaluation. Furthermore, based on the senescence characteristics of litchi fruit at room temperature, the model can be used to determine the objective of senescence regulation according to the characteristics of different varieties and to design personalized fresh-keeping technology and storage methods to extend the postharvest life of fruits as much as possible. Additionally, this study provides new insights into the genetic breeding of litchi germplasm resources. The model can be used to select parameters suitable for breeders, growers, and processors and objectively quantify the quality differences between cultivars with breeding lines. For the different varieties or sources of fruit, researchers only need to measure the relevant characteristic indices during storage and use the predictive model to calculate the browning index and decay rate. The indicators in this study can be used not only as evaluation indicators of litchi storage tolerance but also for litchi breeding patents and the natural hybridization of progeny selection. Additionally, we measured the browning degree and decay rate of 50 litchi varieties and screened out the most storable litchi variety at room temperature, Liu Li 2 Hao, and the more storable litchi varieties Wu Jun 2 Hao, Shang Shu Huai, Qiong Shan 31 Hao, Zu Shan 4 Hao, Jian Ye Li, Jing Xing, Hai Guo 4 Hao, Xue Huai Zi, Jia Yuan Mi Li, Zhuang Yuan 1 Hao, Cheng Tuo, Huai Zhi, and Yuan Duan Li. The practical significance of this study is obvious; in Guangxi, for example, more than 90% of litchi planting is small retail planting. Most fruit farmers do not know how to prolong the storage time of fruit after harvest, leading to the problem of “high yield and slow sales”. Choosing a more storage-tolerant litchi variety can alleviate this phenomenon to some extent. The selection of litchi varieties that are more resistant to storage at normal temperatures and the combination of fresh-keeping treatment technology can further extend the shelf life of fruits and effectively avoid waste.

## 5. Conclusions

The senescence and deterioration of harvested litchi fruits are rapidly reflected on pericarp browning and disease development. The browning index of 50 litchi varieties is significantly decreased and the decay rate significantly increased during storage at room temperature. In addition, storage at room temperature caused a series of changes in the appearance index, physiological index, and quality index of litchi fruit. We screened out the most storage-resistant litchi variety, Liu Li 2 Hao, using principal component analysis and cluster analysis. The stepwise multiple regression analysis further confirmed that relative conductivity, laccase activity, *Vc* content, pericarp thickness, total soluble solids, and *a** and *L** total soluble solids can be used as comprehensive evaluation factors to predict fruit decay rates and browning indices in litchi. The most important significance to emerge from this study is that the characteristic extracted from the liner regression model contributes to simplifying the indicator types, which contribute to predicting the browning index and decay rate by only measuring the appearance-quality-related indicators of fruits during storage. The main strength of this study is that the relevant characteristic indices during storage can be used in the predictive model to calculate the browning index and decay rate and provide guidance for the selection of the varieties of litchi planting and the development of litchi postharvest preservation technology. With regard to the research methods, some limitations need to be acknowledged, namely, the model needs to be further validated and optimized through numerous experiments. Therefore, further studies could evaluate the limitations.

## Figures and Tables

**Figure 1 foods-12-01725-f001:**
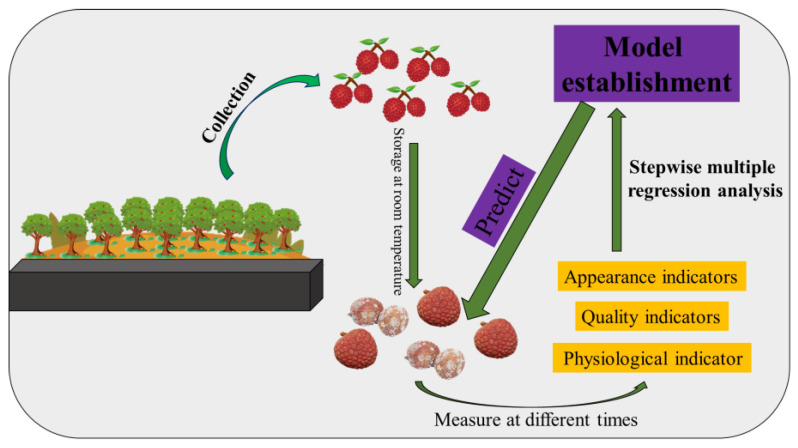
Schematic diagram of the research process.

**Figure 2 foods-12-01725-f002:**
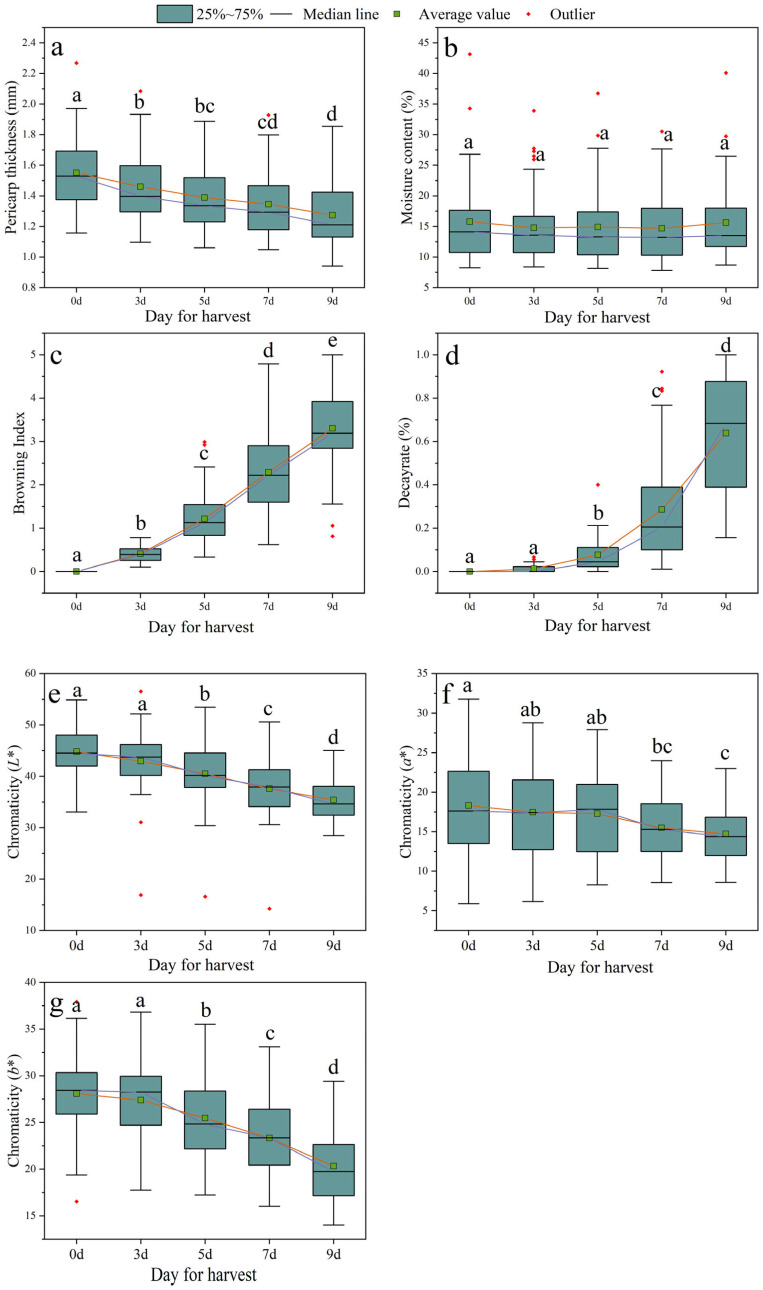
Changes in appearance indices of litchi fruits under normal temperature storage after harvest. (**a**) Pericarp thickness; (**b**) moisture content; (**c**) browning index; (**d**) decay rate; (**e**) *L**; (**f**) *a**; (**g**) *b**. Different letters above groups represent significant differences; shared letters represent no significant differences. The error bars represent standard deviation.

**Figure 3 foods-12-01725-f003:**
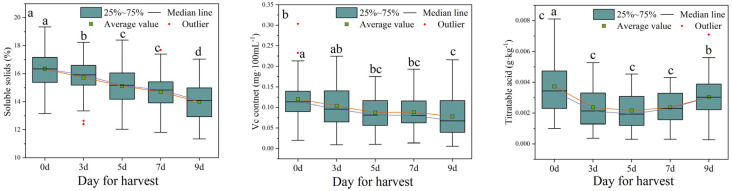
Changes in quality indices of litchi fruits under normal temperature storage after harvest. (**a**) Total soluble solids; (**b**) *Vc* content; (**c**) titratable acid. Different letters above groups represent significant differences; shared letters represent no significant differences. The error bars represent standard deviation.

**Figure 4 foods-12-01725-f004:**
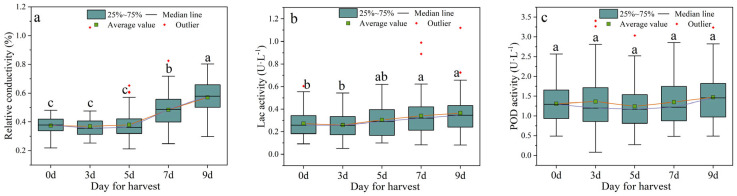
Changes in physiological indices of litchi fruits under normal temperature storage after harvest. (**a**) Relative conductivity; (**b**) laccase activity; (**c**) POD activity. Different letters above groups represent significant differences; shared letters represent no significant differences. The error bars represent standard deviation.

**Figure 5 foods-12-01725-f005:**
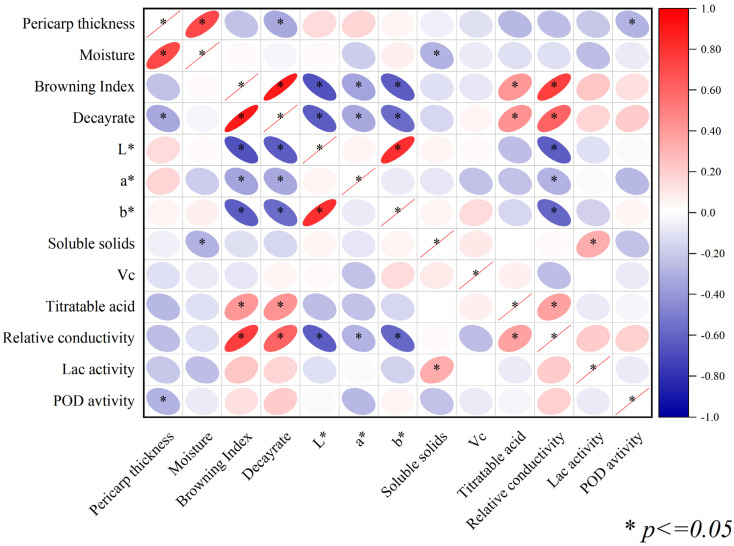
Heat map of correlation among appearance indices, quality indices, and physiological indices. Asterisk represents significance.

**Figure 6 foods-12-01725-f006:**
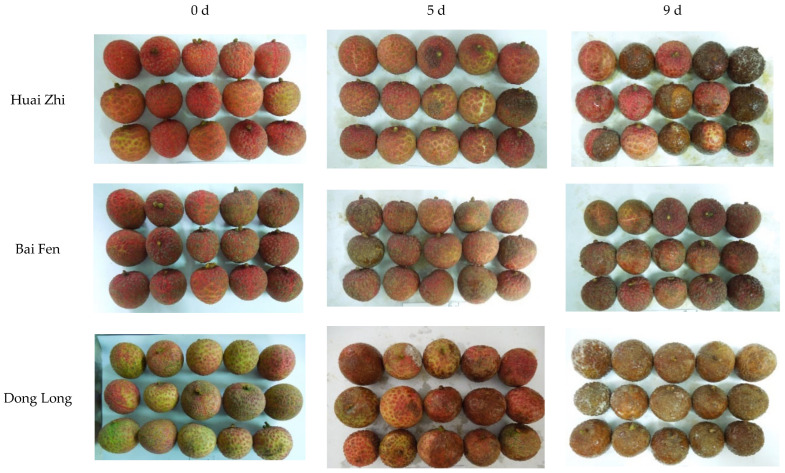
Photos of litchi varieties (part) during normal temperature storage.

**Table 1 foods-12-01725-t001:** The fifty litchi germplasms used for storage.

Maturation Period	Litchi Accession
Early maturing	Bai Tang Ying, Jian Ye Li
Middle maturing	Hei Ye, Yu Tan Mi Li, Huang Pi Li, Wu Cha Li
Middle–late maturing	Xin Qiu Mi Li, Dong Long, Ping Pang Qiu, Chi Tang Li, Chui Li Huang Li, E Dan Li, E Dan Li 2 Hao, Hai Guo 4 Hao, Gui Yuan 1 Hao, Qiong Shan 11 Hao, Liu Li 2 Hao, Xiang Wan, Qiong Shan 31 Hao, An Duo Ji Dan, Cui Ye, Cheng Tuo, Dong Long Mi Li, Gui Wei
Late maturing	Guang Yuan 1 Hao, Xue Huai Zi, Yan Zhi Hong, Hei Zhi, Jiao Pan Li, Ji Cui Rou, Ju Fen 1 Hao, Jia Yuan Mi Li, Yuan Duan Li, Qiong Gong 1 Hao, Chang Hong, Huai Zhi, Zhuang Yuan 1 Hao, Ren Shan Li, Qi Yue Shu, Sha Li, Shang Shu Huai, Bai Fen, Liu Li 1 Hao, Zhu Shan 4 Hao, Qiong Shan 15 Hao, Wu Jun 2 Hao, Yu Lin Li, Ji Gong 1 Hao, Jing Xing, Mu Pai 1 Hao

**Table 2 foods-12-01725-t002:** The load matrix from factor analysis of the relative indices with fruit browning and decay at room temperature.

	Component				
	1	2	3	4	5
Peel thickness	0.74	0.15	−0.035	0.13	0.43
*L** value	0.26	−0.37	−0.27	0.82	−0.19
*a** value	0.43	0.38	−0.64	−0.14	0.12
Total soluble solids	0.65	0.47	0.34	0.005	−0.18
Ascorbic acid	0.48	−0.27	0.7	−0.03	−0.12
relative conductivity	−0.45	0.37	0.38	0.4	0.56
Laccase activity	−0.25	0.77	0.048	0.24	−0.43
Eigen value	1.72	1.32	1.22	0.92	0.78
variance (%)	24.57	18.88	17.49	13.17	11.1
Accumulated variance (%)	24.57	43.45	60.94	74.11	85.21

Note: Loading values above 0.5 were italic values.

**Table 3 foods-12-01725-t003:** Results from the total scores and clustering for the fruit storability of fifty cultivars at room temperature storage.

Litchi Accession	Browning Score	Rotting Score	Storage Stability Classification
Liu Li 2 Hao	10.6	10.6	1
Wu Jun 2 Hao	7.94	7.94	2
Shang Shu Huai	7.91	7.91	2
Qiong Shan 31 Hao	7.89	7.89	2
Zu Shan 4 Hao	7.78	7.78	2
Jian Ye Li	7.73	7.73	2
Jing Xing	7.63	7.63	2
Hai Guo 4 Hao	7.62	7.62	2
Xue Huai Zi	7.58	7.58	2
Jia Yuan Mi Li	7.55	7.55	2
Zhuang Yuan 1 Hao	7.55	7.55	2
Cheng Tuo	7.51	7.51	2
Huai Zhi	7.48	7.48	2
Yuan Duan Li	7.47	7.47	2
Xin Qiu Mi Li	7.4	7.4	3
Gui Yuan 1 Hao	7.39	7.39	3
Bai Tang Ying	7.33	7.33	3
Qiong Shan 11 Hao	7.31	7.31	3
Wu Cha Li	7.31	7.31	3
Hei Zhi	7.29	7.29	3
Ping Pang Qiu	7.29	7.29	3
Sha Li	7.24	7.24	3
Cui Ye	7.24	7.24	3
Qi Yue Shu	7.21	7.21	3
Qiong Gong 1 Hao	7.2	7.2	3
Bai Fen	7.17	7.17	3
Gui Wei	7.14	7.14	3
Ju Fen 1 Hao	7.12	7.12	3
Yu Tuan Mi Li	7.11	7.11	3
Hei Ye	7.08	7.08	3
Chang Hong	7.05	7.05	3
Huang Pi Li	6.92	6.92	4
Guang Yuan 1 Hao	6.9	6.9	4
Liu Li 1 Hao	6.9	6.9	4
Chui Li Huang Li	6.85	6.85	4
Qiong Shan 15 Hao	6.84	6.84	4
Mu Pai 1 Hao	6.84	6.84	4
Yu Lin Li	6.83	6.83	4
Xiang Wan	6.81	6.81	4
Chi Tang Li	6.73	6.73	4
Ji Cui Rou	6.71	6.71	4
E Dan Li	6.63	6.63	4
Yan Zhi Hong	6.56	6.56	4
Ji Gong 1 Hao	6.54	6.54	4
Dong Long	6.49	6.49	4
An Duo Ji Dan	6.39	6.39	5
Ren Shan Li	6.32	6.32	5
E Dan Li 2 Hao	6.29	6.29	5
Jiao Pan Li	6.16	6.16	5
Dong Long Mi Li	6.03	6.03	5

**Table 4 foods-12-01725-t004:** Browning index stepwise regression results.

	Unstandardized Coefficients		Standardized Coefficients	t	Sig.
	B	Std. Error	Beta		
Constant	5.184	0.460		11.269	0.000
Relative leakage rate	4.905	0.399	0.463	12.302	0.000
Total soluble solids	−0.280	0.028	−0.415	−9.923	0.000
Peel thickness	−0.924	0.211	−0.176	−4.386	0.000
Laccase activity	3.250	0.694	0.182	4.681	0.000
*L** value	−0.004	0.002	−0.077	−2.107	0.036
*a** value	−0.026	0.010	−0.099	−2.534	0.012
Ascorbic acid	−2.244	1.041	−0.087	−2.156	0.032

## Data Availability

The datasets generated for this study are available on request to the corresponding author.

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
