# Peer review of "Storability and Linear Regression Models of Pericarp Browning and Decay in Fifty Litchi (Litchi chinensis Sonn.) Cultivars at Room Temperature Storage"

_foods, 2023, doi:10.3390/foods12081725_

Round 1

Reviewer 1 Report

The manuscript entitled “Storability and linear regression models of pericarp browning and decay in fifty litchi (Litchi chinensis Sonn.) cultivars at room temperature storage” is interesting and relevant to the journal’s scope. However, a through revision/justification is required prior to publication.

In abstract section: Summarize the key results. If possible, add the numerical description of most important results.

Introduction section contains various unnecessary statements. Remain focused on the topic. Begin with the broadest scope and get progressively narrower, leading steadily to the statement of objectives. Be clear regarding objectives.

However, there are few reports regarding the application of this approach to litchi?? Add reference

Material and Method section: Clarify the experimental design.

Avoid starting a sentence with number/abbreviation. 

In results, better to add the numeric description of results (% variations) for easy understanding of the readers.

Discussion should be merely based on the observed findings. Not just a review of literature. Answer the question posed in introduction and correlate your finding with the existing knowledge.

Conclusion: Report the key findings… it should be mechanistic. Possible basis of observed results?

Check whether the format of all references is according to the journal format.

Language needs substantial improvement. There are several grammatical and typo mistakes throughout the manuscript.

All the tables and figures should be self explanatory. Define all the abbreviations in the table foot note/figure captions.

In figure captions: specify what do a capped bar and lettering above mean indicate?

Reviewer 2 Report

Dear authors,

The manuscript entitled "Storability and linear regression models of pericarp browning and decay in fifty litchi (Litchi chinensis Sonn.) cultivars at room temperature storage", represents an original and very useful work to face the fact of the damage that could occur during the process. marketing for Litchi chinensis Sonn. taking into account the particular characteristics that the various varieties of this fruit that are marketed may have.

In general, the manuscript is well structured and written, and its introduction adequately presents the problem, the materials and methods, describes the necessary analyzes to be carried out and the results and discussion support the conclusions reached.

Despite the undeniable merits of the manuscript, some easily correctable minor errors and problems should be corrected to improve the quality of the manuscript.

All of them have been pointed out in the attached manuscript itself.

These are:

1) The equation editor of MS Word must be used, and each of its components must be numbered and explained.

2) Some abbreviations are not explained in the text or are explained after being used for the first time.

Greetings,

Reviewer

Reviewer 3 Report

The context should be better described in the introduction, with particular regards related to innovative approach of modeling development and clustering for fruit storability.

Major details of Table 1 should be given and also graphical scheme of study approach should be inserted.

Major details on physiological indicators should be added.

Results on changes of quality indicators should be better described.

Major details on description of heat maps should be given.

In the section 4.3. The application of the linear regression models should be explained better.

Limits, advantages and future directions should be inserted in the Conclusion.

Round 2

Reviewer 1 Report

The authors have addressed my comments; I am happy to recommend acceptance.